# Valorization of Carob and Brewer’s Spent Grain as Growth-Substrate Supplements in *Tenebrio molitor* Rearing

**DOI:** 10.3390/ani15121697

**Published:** 2025-06-08

**Authors:** Irene Ferri, Matilda Rachele Dametti, Sara Frazzini, Matteo Dell’Anno, Luciana Rossi

**Affiliations:** Department of Veterinary Medicine and Animal Sciences, University of Milan, 26900 Lodi, Italy; irene.ferri@unimi.it (I.F.); matilda.dametti@unimi.it (M.R.D.); matteo.dellanno@unimi.it (M.D.); luciana.rossi@unimi.it (L.R.)

**Keywords:** *Tenebrio molitor*, insect meals, carob, brewer’s spent grain, agro-industrial by-products, functional feed, antioxidant activity, total phenolic content, sustainable protein source, circular economy

## Abstract

As the global demand for sustainable protein sources grows, edible insects like *Tenebrio molitor* (yellow mealworm) are gaining attention for their ability to convert food waste into high-quality protein. This study tested whether two common by-products, carob and brewer’s spent grain, could be added to mealworm growth substrate to enhance their nutritional and functional properties. The insects fed with carob and brewer’s spent grain showed better growth and efficiency than those fed with a standard substrate. However, despite the high levels of antioxidants in these by-products, the resulting insect meals had lower antioxidant levels. This may be due to how the insects process these compounds or how their faster growth affects antioxidant balance. These results show that by-products can be used in insect farming to improve sustainability, but their effects on the final product need to be carefully considered. This research contributes to the development of more sustainable and functional animal feed using insect-based ingredients.

## 1. Introduction

The increasing pressure on global food systems due to climate change, depletion of natural resources, and a rapidly growing population has raised significant concerns about the sustainability of conventional food and feed production [1]. Among the most pressing concerns is the projected 52% increase in global demand for animal protein by 2050, according to the FAO [2]. In this scenario, the urgency to explore alternative and sustainable protein sources is driven not only by the rising demand for animal-origin protein but also by the growing environmental and resource constraints associated with conventional production systems [3].

Insects have emerged as a promising solution, offering high nutritional value, efficient feed conversion rates, and a relatively low environmental footprint. In the European Union, several insect species have been approved for human consumption under EC Regulation No. 2015/2283 and for use in animal feed under EC Regulation No. 2017/893 [4,5]. Among them, *Tenebrio molitor*, commonly known as the yellow mealworm, gained increasing attention due to its rich protein and lipid content, as well as its functional properties [6]. Beyond their basic nutritional value, *T. molitor* larvae contain various bioactive compounds, including antimicrobial and antioxidant molecules, making them suitable candidates for functional food and feed production [7,8]. Notably, this species also exhibits a unique ability to bioaccumulate nutrients and phytochemicals from its growth substrate, enabling the valorization of agro-industrial and agricultural by-products in a circular economy framework [9].

In this context, the literature has explored the use of plant-based waste materials as alternative feed ingredients in insect rearing [10]. Carob (*Ceratonia siliqua* L.) and brewer’s spent grain (BSG) are two by-products with high nutritional and functional value. Carob, a leguminous plant widely used in food and cosmetic industries, is rich in carbohydrates, fiber, essential minerals, and vitamins [11,12]. Notably, it contains significant levels of phenolic compounds with recognized antioxidant activity [13,14]. In vivo studies have demonstrated carob’s potential to promote body weight gain, improve α-tocopherol levels, and positively modulate immune function in livestock animals [15,16,17]. Its use as a rearing substrate for *T. molitor* has been associated with increased phenolic content and enhanced antioxidant activity in the larvae [18].

Similarly, BSG—a major by-product of the brewing industry composed primarily of barley husks and seed coat residues—is an abundant source of protein, fiber, minerals, and phenolic compounds [19,20]. These include ferulic, syringic, and p-coumaric acids, all of which exhibit significant antioxidant effects [21]. Hydrolyzed BSG has also been shown to generate antioxidant peptides with radical scavenging capacity [22]. In animal nutrition, BSG supplementation has led to improved growth performance, immune responses, and antioxidant capacity, without detrimental effects on metabolic profiles [23,24,25]. Studies applying BSG to *T. molitor* rearing have demonstrated enhanced larval growth, shorter development cycles, and improved protein and fat contents [26,27].

These findings suggest that the nutritional profile and functional quality of insect meals can be modulated by the inclusion of selected by-products in the growth substrate. However, while previous studies have largely focused on growth performance and basic nutritional outcomes, fewer investigations have addressed the influence of such substrates on the functional properties of the resulting insect meals—particularly in terms of total polyphenol content and antioxidant capacity, both of which are essential for evaluating their value in functional feed applications.

Therefore, the aim of the present study was to evaluate the effects of supplementing standard wheat bran substrate with carob or BSG on growth performance and nutritional and functional quality of *T. molitor* insect meals. Specifically, we investigated the total polyphenol content and antioxidant activity (both water- and lipid-soluble fractions) of the resulting larval meals to better understand the potential of these by-products in enhancing the biofunctional profiles of insect-based ingredients for sustainable nutrition and circular economy strategies.

## 2. Materials and Methods

### 2.1. Experimental Design

In the present study, larvae from the time of birth for the next 7 weeks were reared on a wheat bran substrate to which vegetables (potatoes and carrots) were added to ensure hydration of the larvae. By reaching 7 weeks of age, when the larvae are particularly capable of assimilating functional compounds from the substrate, as suggested by recent studies in the literature [28,29,30], a total of 2.4 kg of *T. molitor* larvae were randomly distributed across 24 plastic containers (27 × 39 × 14 cm), each containing 100 g of larvae (approximately 500 larvae/plastic trays). Larvae were divided into three groups with eight replicates per group based on the growth substrate. The control group (Ctrl, n = 8 trays) was reared on wheat bran, treatment group 1 (Trt1, n = 8 trays) received wheat bran supplemented with 25% carob, and treatment group 2 (Trt2, n = 8 trays) received wheat bran supplemented with 25% BSG. All groups were sprayed with water on days 0, 2, 4, 6, 8, 10, and 12 (10 mL/day). Wheat bran was sourced from a local farm, while carob was supplied as a by-product of the agro-industrial carob production chain. BSG, was provided fresh and wet by the artisanal brewery Soralamà (Vaie, Turin, Italy) immediately after the mashing process. The material did not include hot trub or undergo any preservation treatments, such as silaging, pressing, or drying. Carob and BSG were dried at 120 °C for 135 min until reaching a final moisture content of 5% checked with a moisture analyzer. After drying, they were ground using a mill equipped with a 2 mm grid screen, to match the particle size of the wheat bran used as the control substrate.

Therefore, larvae were reared for 14 days under controlled environmental conditions (26 ± 2 °C, 60 ± 5% relative humidity) at the Italian Cricket Farm s.r.l. (Pinerolo, Italy), allowing them to reach the final stage of the larval phase, before pupation. Throughout the rearing period, daily monitoring was performed to ensure the consistency of the experimental conditions.

At each weekly sampling timepoint, the entire growth substrate from each replicate was weighed to assess consumption. After weighing the residual biomass, representative samples were taken on days 0 and 14 and stored at −20 °C for further analysis. Fresh substrate was then added to maintain a 2:1 larvae-to-substrate ratio. The larvae weight per container was recorded every 7 days. On day 14, all larvae from each tray were manually separated from the substrate using a 300 µm mesh sieve and collectively weighed to determine the total fresh biomass using a precision scale (B2002-S, Mettler Toledo, Milan, Italy, accuracy = 0.01 g) [31]. After a 24 h fasting period, the larvae from each container were cooked by microwave drying (model CMG2071M, Candy Hoover Group S.r.l., Brugherio, Italy) at a maximum input power of 120 W and a frequency of 2450 MHz for 5 min, following the microwave cooking method described by Mancini et al. [32]. The dried larvae were then ground into meal using a flour mill, resulting in eight replicates per group for the subsequent laboratory analysis.

### 2.2. Tenebrio Molitor Larvae Performance

The residual substrate weight (grams) was recorded after separation from the larvae on days 7 and 14. Substrates were sifted using a 150 μm mesh sieve to separate feces from non-ingested feed. The feed consumption (%) was calculated as the difference between the initially provided substrate and the remaining feed at each time point. To evaluate the feed conversion efficiency, the feed conversion ratio (FCR) was determined on a dry matter basis by dividing the weight of ingested feed by the weight gain of the larvae. Larval mortality was recorded weekly alongside the separation of insects and residual substrates. The survival rate was calculated as the percentage of live larvae at the end of the trial. At the beginning of the trial, the number of larvae was estimated by dividing the total larval biomass by the average individual weight of 7-week-old *T. molitor* larvae (~0.2 g). Mortalities were recorded manually by counting dead individuals in each tray on days 7 and 14. The weight of the deceased larvae was subtracted from the total biomass to estimate the number of surviving larvae at day 14, which was then used to calculate the survival rate. While this method provides an indirect estimation, it was adopted due to the experimental setup and feasibility constraints. Additionally, the growth rate of *T. molitor* was evaluated according to the following formula:$$Growth\;rate\;\left(\%\right)=\left(\frac{Final\;weight-Initial\;weight}{Initial\;weight}\right)\times 100$$

### 2.3. Chemical Characterization of Rearing Substrate

The chemical composition of the substrates, represented by the input feed sampled on day 0 and after 14 days, was assessed, following the “Official Methods of Analysis” protocols [33], after milling through a 1 mm screen. This mesh size was selected to ensure compatibility with the size of the larvae and to effectively separate most frass and fine residues. Although minimal contamination with fine particulate matter cannot be completely excluded, the sieving procedure was performed carefully and consistently to minimize this risk. Dry matter content was determined by drying the substrate samples at 65 °C for 24 h (AOAC 930.15). The lipid content (ether extract, EE) was measured using ethyl ether in a Soxtec extractor (VELP Scientifica Srl, Usmate, Italy) (AOAC 2003.05). The total ash content was obtained by incinerating samples at 550 °C for 3 h (AOAC 942.05). Crude protein (CP) levels were analyzed using the Kjeldahl method, applying a nitrogen conversion factor of 6.25 for plant-based substrates (AOAC 2001.11). Crude fiber (CF) was determined using AOCS Ba 6a-05 methods with filtering bags. Non-structural carbohydrates (NSCs) were calculated as follows [34]:$$NCS=[\;100-\left(moisture\%+ash\%+EE\%+CP\%+CF\%\right)]$$

Each batch of rearing substrate was analyzed independently, with thorough homogenization to ensure representativeness. Substrates from each group were tested as independent replicates (n = 8), and each analysis was performed in technical triplicate, resulting in 24 total determinations.

### 2.4. Chemical Characterization of Tenebrio molitor Larvae Meal

The chemical composition of the *T. molitor* larvae meal, represented by the processed larvae, was assessed as described in the previous section. For the determination of the CP, a nitrogen conversion factor of 4.76 was used according to Janssen et al. (2017) [35]. Each batch of insect meal was analyzed separately, with independent replicates for each group (n = 8), and each sample was analyzed in technical duplicates to ensure analytical reliability.

### 2.5. Extraction of Tenebrio molitor Larvae Meal

The lipid-soluble and water-soluble fractions of the insect meals were extracted following the method described by Di Mattia et al. (2019), with some modifications [36]. Briefly, 4 g of insect meals were weighed and defatted. The defatting process was performed through three washing cycles with 25 mL of hexane, each involving 1 min of vortexing followed by 10 min of centrifugation at 5000 rpm and 4 °C. The resulting hexane fractions were pooled to obtain the lipid-soluble extract. The residual solid fraction was air-dried to completely remove the hexane and considered as the defatted insect meal.

For the water-soluble fraction, 1 g of dried and defatted insect meals was mixed with 25 mL of distilled water to form a homogenized mixture. The mixture was vortexed for 1 min to ensure uniform consistency. It was then transferred to a 50 mL vial, which was wrapped in aluminum foil to protect it from light and placed in a shaker for 1 h at 18 °C under dark conditions. After extraction, the mixture was centrifuged at 5000 rpm for 15 min at 4 °C to separate solid residues from the liquid extract. The supernatant was filtered through cellulose filters to remove the remaining particulates. Finally, the volume of the filtered extract was adjusted to 25 mL with distilled water.

Both extraction procedures, for the lipid-soluble and water-soluble fractions, were performed in biological triplicate.

### 2.6. Evaluation of Total Phenolic Content (TPC) of Tenebrio molitor Larvae Meal Extracts

The TPC of the insect meal extracts, at different dilutions (100%; 50%; 25%; 12.5%; 6.25%; 3.12%), was evaluated by the Folin–Ciocalteu method, according to Attard et al. (2013) [37]. The TPC was determined spectrophotometrically at 630 nm (BioTek Epoch Microplate Sepctrophotometer, Agilent Technologies, Santa Clara, CA, USA). Calibration curves were prepared in five 1:2 dilutions, from 960 μg/mL to 60 μg/mL, with tannic acid as the standard. Each sample and standard were performed in triplicate. The TPC was expressed as mg of tannic acid equivalents (TAE) per 100 g of insect meals (mg TAE/100 g).

### 2.7. Evaluation of Antioxidant Properties of Tenebrio molitor Larvae Meal Extracts

The antioxidant activity of *Tenebrio molitor* larvae meals was assessed using the ABTS assay, following the methodology outlined in our previous study [38]. For the water-soluble fraction, 10 µL of the appropriately diluted sample was added to 1 mL of the ABTS•^+^ working solution in water adjusted to an absorbance of 0.70 ± 0.02 nm. After 6 min under dark conditions, the absorbance was measured. For the lipid-soluble fraction, 180 µL of the sample was mixed with 2.82 mL of the ABTS•^+^ solution prepared in ethanol (Abs 0.70 ± 0.02 nm), and the absorbance was measured after 1 h under dark conditions, as reported by Sacchetti et al. (2008) [39]. Two calibration curves were generated for the water- and lipid-soluble fractions using Trolox (6-hydroxy-2,5,7,8-tetramethylchroman-2-carboxylic acid) as the standard, prepared in water and ethanol, respectively. Trolox was tested at concentrations of 2000 µM, 1500 µM, 1000 µM, 500 µM, 250 µM, and 0 µM. All tests were conducted in triplicate.

### 2.8. Statistical Analysis

Statistical analyses were conducted using GraphPad Prism (Version 9.1.1). For insect growth, a generalized linear model was employed after assessing the normality and homoscedasticity through Q–Q plots and tests (Shapiro–Wilk test for normality and Bartlett’s test for homoscedasticity). The model considered the fixed effects of treatment, time (day or hour), and their interaction. Chemical composition, total phenolic content, and antioxidant activities were analyzed via one-way ANOVA, following verification of normality (Shapiro–Wilk test) and homoscedasticity (Bartlett’s test). Mortality was assessed by calculating the survival rate averaging the number of dead larvae over the 14-day period, with the group means of the unpaired samples compared using ANOVA. The content of polyphenols and the antioxidant activity of the insect meal extract were analyzed through two-way ANOVA based on a full factorial model, where treatment and time were included as fixed effects, and the tray was included as a random effect to account for variability among the experimental units. The resulting data are described as the mean ± standard deviations. The statistical significance was set at *p* < 0.05.

## 3. Results and Discussion

### 3.1. Growth Performance, Feed Conversion Efficiency, and Survival Rate of Insects

The substrate was consumed equally by all experimental groups during the entire test period (Table 1).

Significant differences in the average weight, growth rate, and feed conversion ratio (FCR) were observed between the control group (Ctrl) and the groups supplemented with carob (Trt1) and BGS (Trt2).

Survival rates remained comparable among the groups, demonstrating high percentages across all treatments (Figure 1).

The incorporation of carob and BSG into mealworm diets has significant implications for their nutritional composition and overall growth performance, highlighting the remarkable bioconversion capabilities of insects. In fact, the increases in the weight and growth rate observed in both treatment groups could be explained by the composition of the carob and brewer. Carob, a leguminous pod, is rich in fibrous material that is crucial for improving digestive health, which is essential for optimal growth performance [18]. Research indicates that mealworms exhibit a favorable growth performance when carob is included in their diet. The presence of carob can lead to increased weight gain and size, although the specific effects may vary depending on the proportion of carob used [40]. Although in this study the sugar composition of carob was not directly analyzed, its high content of soluble sugars—mainly sucrose, glucose, and fructose—is well documented in the literature, with concentrations reaching up to 50–55% of the dry weight [11,12]. However, considering the inclusion level of carob in our experimental substrate (25%), the total sugar intake remained within a range that is unlikely to impair mealworm digestion or metabolic activity. Insects like *T. molitor* are capable of metabolizing moderate levels of dietary sugars efficiently, and previous studies have shown that inclusion rates at or below this level do not negatively affect larval growth or nutrient assimilation [41]. On the contrary, such inclusion may provide an additional energy source that supports biomass accumulation. Therefore, while a detailed sugar analysis would further refine these insights, the current formulation appears appropriate and effective for mealworm development without evidence of adverse effects. Similarly, BSG, although an agro-industrial by-product, effectively supported larval growth, thanks to their contents of dietary fiber and protein. Studies indicate that incorporating BSG into mealworm diets can lead to increased growth rates and feed conversion efficiency, making it an attractive option for sustainable insect farming [42,43,44]. Obtaining a lower FCR in the treated groups compared with the control group indicates that both carob and BSG demonstrated a high conversion efficiency compared with bran alone. In fact, a lower FCR value means that a smaller amount of feed is required to produce the same amount of biomass, reflecting a more efficient nutrient assimilation and growth. Therefore, the data obtained in this study further suggest the potential of locust beans and BSG as sustainable and nutritionally effective ingredients for insect farming. A study conducted by Yu et al. (2024) demonstrated that mealworms fed with BSG had a robust growth performance, exhibiting a survival rate of 98.33% and a biomass increase of 23.06% compared with other industrial food wastes [45]. As well, Mancini et al. (2022) highlighted superior growth rates of *T. molitor* reared on diets with BSG compared to those fed with bread leftovers [27]. Moreover, in line with the results obtained in this study, previous research by Kim and colleagues reported an optimal growth rate in mealworms fed with a diet containing 30–50% BSG mixed with wheat bran, which resulted in higher larval weight and pupation rates [26]. However, it is important to note that the effect of BSG on the growth performance of *T. molitor* is also related to the administered form used. When BSG was used in its wet form, an impaired growth performance in mealworms was observed, likely due to excess moisture or microbial activity affecting substrate quality. For instance, Deruytter and colleagues observed that uneven distribution of wet feed, such as agar used as a proxy for moist substrates, led to decreased larval density and growth rates [46]. These findings suggest that both the physical form and distribution of wet feed components like BSG play a critical role in determining their suitability as feed components.

### 3.2. Chemical Composition of Rearing Substrates

The chemical analysis of the rearing substrates revealed distinct differences in the macronutrient compositions depending on the type of supplementation used. Although no significant variations were observed in the dry matter, ash, or ether extract contents among the groups, notable differences emerged in the levels of crude fiber, crude protein, and non-structural carbohydrates (NSCs) (Table 2).

The Trt1 substrate, which included 25% carob, exhibited the highest crude fiber content (18.70 ± 1.02%) compared to the control (11.87 ± 4.30%) and Trt2 (16.31 ± 0.48%) groups, with a statistically significant difference (*p* = 0.0446). This result is consistent with the existing literature, highlighting carob’s high dietary fiber content, particularly insoluble fiber derived from its pod matrix [11]. Increased dietary fiber may influence larval digestion and gut motility, potentially impacting nutrient absorption and feed conversion efficiency [47,48].

Crude protein levels also varied significantly across groups (*p* = 0.0107), with Trt2 (BSG-enriched) presenting the highest protein content (18.14 ± 0.90%) and Trt1 the lowest (15.92 ± 0.24%). This aligns with previous findings that BSG is a protein-rich by-product, often used to enhance protein levels in animal feed [20]. Conversely, the moderate protein content in the carob-based diets may explain the comparatively lower protein level in Trt1.

Additionally, the NSC content was significantly lower in the supplemented groups compared with the control (*p* = 0.0242), likely due to the partial replacement of carbohydrate-rich wheat bran with fibrous by-products. As NSCs are a primary energy source, their reduction could influence larval energy metabolism depending on the digestibility of alternative nutrients introduced through the supplements.

### 3.3. Chemical Composition of Tenebrio molitor Meals

The nutrient composition of the experimental *Tenebrio molitor* larvae was analyzed in this study, as diet significantly impacts their proximate composition. A comparable chemical composition was observed among the experimental groups, except for dry matter content, which was highest in the control group (Ctrl). This variation may reflect minor differences in moisture loss during microwave baking, although the same thermal treatment protocol was applied across all groups. Despite significant differences in protein and fiber contents in the growth substrates, the resulting larval meals exhibited similar protein concentrations (Table 3). This observation aligns with the well-documented ability of *T. molitor* to physiologically buffer dietary variability through efficient bioconversion mechanisms. Insects such as *T. molitor* can reallocate available nutrients to maintain a relatively stable body composition, particularly in terms of protein and lipid contents, despite variations in substrate quality. This metabolic adaptability likely contributes to the consistent nutritional profile of the larvae observed across treatments [49].

It is worth noting that the chemical compositions of the substrates were analyzed at different time points (day 0 and day 14), while the larval meal composition was assessed only at the end of the trial. Although this temporal discrepancy does not allow for a direct comparison, the observed changes in substrate composition suggest that nutrient availability evolved during the rearing period and may have influenced larval performance and conversion efficiency. However, the results do not reveal marked differences in the final larval protein contents, supporting the idea of physiological homeostasis in nutrient allocation.

The lack of significant differences in the protein contents of the insect meals belonging to the different experimental groups may be due to the inclusion rates used. In this study, both carob and brewer’s spent grain were incorporated at a 25% level in the rearing substrate. Although this proportion was sufficient to influence feed conversion and growth performance, it may not have been high enough to cause detectable changes in larval protein content. Previous research has indicated that more pronounced effects on the protein composition of *Tenebrio molitor* may occur at higher inclusion levels (≥40%), particularly when using protein- or energy-dense feed components [41,50]. These findings suggest that the nutritional response of insects to dietary modifications is both ingredient- and dose-dependent. In line with this, Couto et al. reported that even if the presence of carob can lead to increased weight gain and size, the specific effects may vary depending on the proportion of carob used [40]. As also revealed by the chemical analysis performed on the growth substrate, BSG is a nutritionally dense by-product containing approximately 25% protein and 47% fiber, among other essential nutrients [51]. BSG’s high protein content makes it an ideal candidate for use as a feed substrate in insect farming, particularly for *T. molitor* larvae [52]. In this context, different studies highlight that the inclusion of BGS in the growth substrate of *T. molitor* leads to a significant increase in the protein content of mealworm meals [27,45]. Despite this, the inclusion of BGS, also in relation to the inclusion level of the by-product, could influence the gut microbiota of mealworms, which plays a vital role in their digestion and nutrient absorption [53]. This could explain our results, which show no statistical difference in the protein contents between the control and the experimental group reared with the BGS inclusion. However, this aspect will require further investigations into the insects’ microbiome. The protein content disclosed in our study is in line with those reported in other studies [54], highlighting the importance of using by-products in insect farming to reduce waste and promote a sustainable economy.

### 3.4. Total Phenol Content (TPC) of Tenebrio molitor Meals

The results obtained demonstrate that the insect diet significantly influences the total phenolic content of the derived meals. Specifically, the water-soluble fraction (Figure 2a) showed a markedly higher phenolic content in the control group compared to the treated groups. These reductions in Trt1 and Trt2 were statistically significant (*p* = 0.0041 and *p* = 0.042, respectively) and may indicate that the inclusion of carob or BSG plays a role in the biosynthesis, accumulation, or extractability of water-soluble phenolic compounds in insects.

A similar trend was observed in the lipid-soluble fraction (Figure 2b), where both treated groups also showed lower total phenolic contents compared with the control. Notably, the most pronounced reduction (*p* = 0.0219) was observed in the BSG group (Trt2), suggesting that this dietary component may more strongly affect the presence of lipophilic phenolic compounds. These findings are particularly relevant in the context of the growing interest in edible insects as a sustainable source of nutrients and bioactive compounds. Polyphenols are known for their antioxidant, anti-inflammatory, and antimicrobial properties, contributing to the functional value of insect-derived ingredients [36,55]. The dietary modulation of insect composition has been reported in the literature, where substrates rich in fiber or polyphenols can influence insect growth performance, gut microbiota, and, ultimately, their metabolic profile [41].

Interestingly, although both carob and BSG are known to be rich in polyphenols themselves, their addition to insect feed did not enhance the polyphenolic content of the resulting meals. This counterintuitive result may be due to several factors, as follows: bioavailability of polyphenols in the feed matrix, insect’s digestive enzymatic capacity, or potential presence of compounds that inhibit polyphenol absorption or metabolism [14,56,57]. It is also possible that the added substrates may have modulated oxidative stress or detoxification pathways in a way that reduced the retention of phenolic compounds, which could account for the lower phenolic content observed in the treated groups relative to the control [54,58].

### 3.5. Antioxidant Properties of Tenebrio molitor Meals

The antioxidant activities of the insect meal extracts, as shown in Figure 3, reveal distinct trends between the water-soluble (Figure 3a) and lipid-soluble (Figure 3b) fractions in response to dietary modifications. In both cases, the control group exhibited the highest antioxidant capacity, followed by Trt1 and Trt2. Significant differences were observed between the Ctrl and Trt1 in the water-soluble fraction (*p* = 0.0172) and between the Ctrl and Trt2 in both fractions (*p* < 0.01), and a modest yet significant decrease was noted between Trt1 and Trt2 in the lipid-soluble extract (*p* < 0.05).

The obtained results firstly suggest that the lipid-soluble fraction showed a content of antioxidant activity lower than the water-soluble fraction, indicating that, regardless of the supplementation, the growing substrate could interfere more significantly with the absorption or transformation of lipophilic antioxidant compounds. These may include tocopherols, carotenoids, and lipid-soluble phenolics, whose bioaccessibility can be hindered by the presence of fibrous- or polyphenol-rich matrices [59,60].

The inclusion of carob or BSG in the insect substrate may reduce the antioxidant potential of the resulting insect meals, particularly in the water-soluble extracted fraction. Consistent with our findings, Antonopoulou et al. (2022) demonstrated that the inclusion of carob in growth substrates, at concentrations similar to those used in this study, resulted in lower antioxidant activity compared to both the control and higher inclusion levels, with the highest antioxidant activity observed at a 75% inclusion of carob [18]. The obtained results could seem somewhat unexpected, as both carob and spent grain are well-documented sources of antioxidant phenolics and other bioactive compounds [19]. However, the antioxidant content of insect-derived extracts is not solely determined by the composition of the feed but is also shaped by the insect’s digestive metabolism, bioavailability of dietary components, and possible interactions during digestion and assimilation [57]. Moreover, high-molecular-weight tannins from the carob or the lignin-rich structure of the spent grain could bind or sequester antioxidants, limiting their incorporation into insect tissues and, consequently, into the lipid fraction of the extract. These findings align well with the total phenolic content measured in the corresponding water and lipid fractions, indicating a close relationship between phenolic content and antioxidant capacity [36]. Further research, for instance, using marked compounds or in vitro digestion models, could help clarify the fate and availability of antioxidant compounds during insect digestion and assimilation.

In addition, it is important to consider that the larvae in the treatment groups (Trt1 and Trt2) exhibited significantly higher weight gains compared with the control (Ctrl) group during the second week of the experiment. Therefore, as the antioxidant activity is expressed in relation to the weight, this could explain the results obtained evaluating the antioxidant capacity of mealworm meal. Specifically, our findings may be attributed to increased metabolic rates in the treatment groups (Trt1 and Trt2), potentially resulting in higher production of reactive oxygen species (ROS), which reduce the overall antioxidant activity. Notably, while antioxidant activity was measured relative to larval weight, it is plausible that the absolute antioxidant activity per individual larva was similar across groups. The heavier body mass of the treatment group larvae could have led to a perceived dilution of antioxidant activity in their extracts.

## 4. Conclusions

In a global context, where sustainability, waste reduction, and functional nutrition are increasingly prioritized, the valorization of agro-industrial by-products through insect rearing represents a promising and multifaced strategy. This study demonstrated that the inclusion of carob and BSG in *Tenebrio molitor* rearing substrates can positively influence larval growth performance and feed conversion efficiency without negatively impacting survival rates. Despite compositional differences in the substrates, particularly in protein and fiber contents, the nutritional profiles of the resulting insect meals remained largely comparable. Interestingly, although both carob and BSG are known for their high polyphenol content and antioxidant potential, their inclusion at 25% resulted in significant reductions in both total phenolic content and antioxidant activity, especially in the water-soluble fractions. This suggests a complex interplay between dietary bioactive compounds, their bioavailability, and the metabolic processes of the insect, highlighting the need to carefully balance the inclusion levels of the functional feed components to optimize both nutritional and biofunctional characteristics of the final meals.

Beyond the nutritional findings, these results also carry important ecological and economic implications. The use of agro-industrial by-products such as BSG and carob pulp in insect farming could contribute to the reduction in waste streams and supports circular economy frameworks. From an industrial perspective, both BSG and carob are already produced on a large scale and are geographically widespread, which enhances their availability and feasibility for standardized inclusion in insect feed formulations. Their integration into insect farming systems could reduce reliance on conventional, more resource-intensive feed ingredients, potentially lowering production costs and the environmental footprint. However, logistical aspects, such as the variability in the by-product composition, storage stability, and local sourcing infrastructure, must be considered to ensure consistency and scalability.

Overall, these findings reinforce the potential of *T. molitor* for bioconverting agro-industrial by-products into high-value protein sources and functional ingredients, supporting sustainable feed development and circular economy strategies. Further studies are needed to evaluate different inclusion levels, a wider range of bioactive-rich by-products, and their long-term effects on insect physiology, product quality, and economic viability under industrial rearing conditions.

## Figures and Tables

**Figure 1 animals-15-01697-f001:**
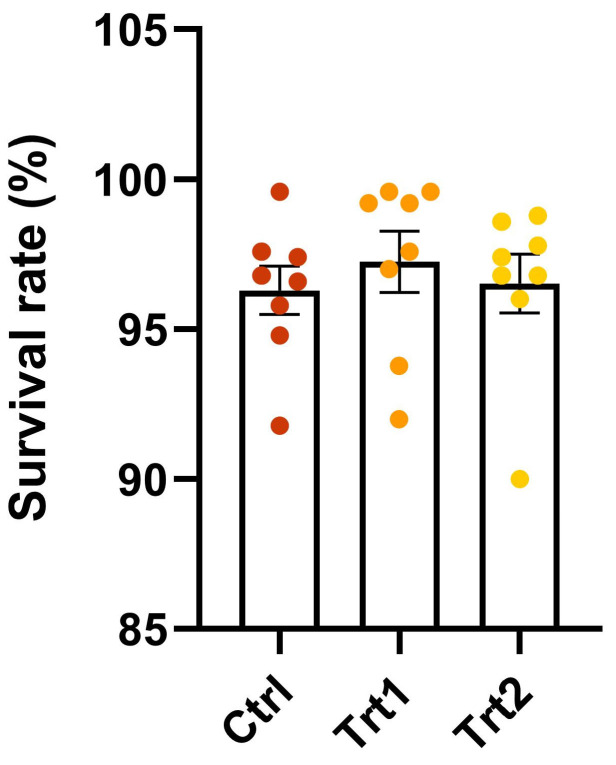
Survival rates of the larvae. The control group (Ctrl) received wheat bran, treatment 1 (Trt1) was fed a substrate constituted of wheat bran supplemented with 25% carob, and treatment 2 (Trt2) was fed a substrate formulated with wheat bran supplemented with 25% brewer’s spent grain (BSG). The values are expressed as the mean ± standard error.

**Figure 2 animals-15-01697-f002:**
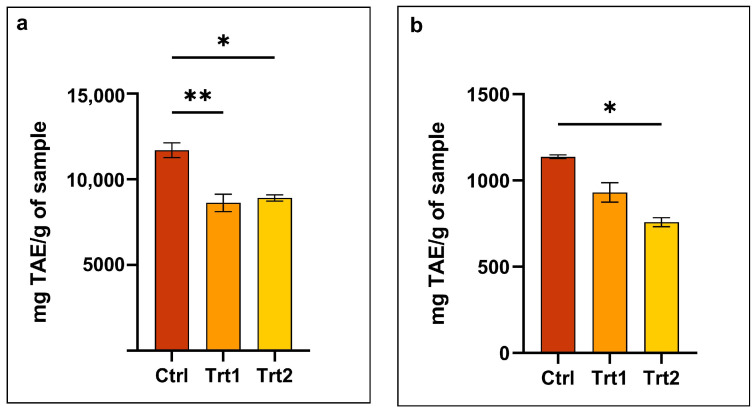
Total phenol content (TPC) in Tenebrio molitor meals: (**a**) water-soluble fraction; (**b**) lipid-soluble fraction. The values are expressed as the mean ± standard error. Asterisks with different superscripts indicate significantly different means, as follows: * *p* ≤ 0.0425 and ** *p* = 0.0041.

**Figure 3 animals-15-01697-f003:**
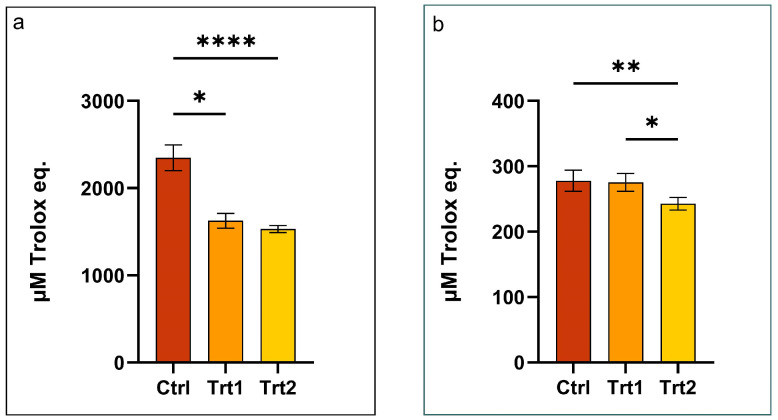
Total phenol content (TPC) in Tenebrio molitor meals: (**a**) water-soluble fraction; (**b**) lipid-soluble fraction. The values are expressed as the mean ± standard error. Asterisks with different superscripts indicate significantly different means, as follows: * *p* ≤ 0.0283; ** *p* = 0.0035; and **** *p* ≤ 0.0001.

**Table 1 animals-15-01697-t001:** Growth performance of *T. molitor* larvae reared on different substrates.

				*p*-Values				
Item	Ctrl	Trt1	Trt2	Trt	Time	Trt x Time	F-Value	DF (n, d)
Substrate consumed (%)				0.0487	0.0897	0.7405	0.30	2, 21
d 0–7	69.82 ± 1.23	70.65 ± 0.36	71.61 ± 0.44					
d 7–14	68.72 ± 0.27	70.29 ± 0.15	70.56 ± 0.09					
Average weight (g)				0.0006	<0.0001	<0.0001	9.44	4, 42
d 0–7	104.5 ± 1.44	107.7 ± 0.85	107.3 ± 0.67					
d 7–14	104.9 ± 2.69 ^a^	114.8 ± 1.26 ^b^	116.9 ± 1.07 ^b^					
Growth rate (%)				0.0005	0.0014	0.0395	3.84	2, 19
d 0–7	4.46 ± 1.45	7.72 ± 0.86	7.13 ± 0.66					
d 7–14	4.05 ± 0.17 ^a^	12.11 ± 1.44 ^b^	12.10 ± 1.42 ^b^					
FCR				0.0068	0.0193	0.752	0.34	2, 18
d 0–7	9.16 ± 2.25	6.16 ± 0.60	6.48 ± 0.58					
d 7–14	8.36 ± 0.40 ^a^	3.97 ± 0.56 ^b^	3.97 ± 0.64 ^b^					

The control group (Ctrl) received wheat bran, treatment 1 (Trt1) was fed a substrate constituted of wheat bran supplemented with 25% carob, and treatment 2 (Trt2) was fed a substrate formulated with wheat bran supplemented with 25% brewer’s spent grain (BSG). Values are expressed as the mean ± standard error. ^a,b^ Different subscripts indicate statistically significant differences among the tested groups (ordinary one-way ANOVA, *p* < 0.05).

**Table 2 animals-15-01697-t002:** Chemical composition of different growth substrates.

Components (%)	Ctrl	Trt1	Trt2	*p*-Values
**Dry Matter**	90.62 ± 1.26	89.10 ± 0.15	89.42 ± 0.18	0.0997
**Ash**	6.09 ± 1.86	7.40 ± 0.32	7.15 ± 0.29	0.3625
**Ether Extract**	2.93 ± 0.94	2.57 ± 0.32	2.17 ± 0.30	0.3602
**Crude Fiber**	11.87 ± 4.30 ^a^	18.70 ± 1.02 ^b^	16.31 ± 0.48 ^ab^	0.0446
**Crude Protein**	17.00 ± 0.42 ^ab^	15.92 ± 0.24 ^a^	18.14 ± 0.90 ^b^	0.0107
**Non-Structural Carbohydrates**	62.11 ± 3.77 ^a^	55.40 ± 1.11 ^b^	56.22 ± 0.96 ^b^	0.0242

The control group (Ctrl) received wheat bran, treatment 1 (Trt1) was fed a substrate constituted of wheat bran supplemented with 25% carob, and treatment 2 (Trt2) was fed a substrate formulated with wheat bran supplemented with 25% brewer’s spent grain (BSG). Ash, ether extract, crude fiber, crude protein, and non-structural carbohydrates are expressed as percentages on a dry matter basis and are presented as the mean ± standard error. ^a,b^ Different subscripts indicate statistically significant differences among the tested groups (ordinary one-way ANOVA, *p* < 0.05).

**Table 3 animals-15-01697-t003:** Chemical compositions of the *T. molitor* larvae meals reared on different substrates.

Components (%)	Ctrl	Trt1	Trt2	*p*-Value
**Dry matter**	94.23 ± 1.84 ^a^	83.42 ± 1.75 ^b^	85.13 ± 3.44 ^b^	0.0036
**Crude protein**	56.84 ± 1.68	58.57 ± 1.56	56.99 ± 2.33	0.7734
**Ether Extract**	27.60 ± 2.30	25.23 ± 1.62	25.33 ± 3.30	0.4383
**Ash**	3.63 ± 0.47	3.81 ± 0.48	3.98 ± 0.32	0.6326

The control group (Ctrl) received wheat bran, treatment 1 (Trt1) was fed a substrate constituted of wheat bran supplemented with 25% carob, and treatment 2 (Trt2) was fed a substrate formulated with wheat bran supplemented with 25% brewer’s spent grain (BSG). Crude protein, ether extract, and ash are expressed as percentages on a dry matter basis and are presented as the mean ± standard error. ^a,b^ Different subscripts indicate statistically significant differences among the tested groups (ordinary one-way ANOVA, *p* < 0.05).

## Data Availability

All data are available within the manuscript and from the corresponding author upon reasonable request.

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
