# Peer review of "Valorization of Carob and Brewer’s Spent Grain as Growth-Substrate Supplements in Tenebrio molitor Rearing"

_animals, 2025, doi:10.3390/ani15121697_

Round 1
Reviewer 1 Report
Comments and Suggestions for Authors
The study is significant in innovation because the effect of using carob and brewer's spent grain as growth substrate supplements in Tenebrio molitor rearing has not been explored well, and data on their use are scarce. The research results provide valuable insight into their potential use as substrates for insect rearing.
A few issues in the manuscript need to be addressed:
The titles of Tables 1, 2, and 3 are too long and need to be shortened. The part explaining tretaments and the statistical methods used should be placed below the table “control group (Ctrl) received wheat bran, treatment 1 (Trt1) fed substrate constituted of wheat bran supplemented with 25% of carob, treatment 2 (Trt2) fed substrate formulated with wheat bran supplemented with 25% of brewer's spent grain. Values are expressed as mean ± standard error. a,b Different subscripts indicate statistically significant differences between tested groups (Ordinary one-way ANOVA p<0.05).”
In some parts of the manuscript, the full name for BSG is used, while in others the abbreviation appears. It is necessary to ensure consistency throughout the entire manuscript.
The primary concern relates to the short duration of the insect rearing trial. The authors reared Tenebrio molitor for only 14 days, which is insufficient according to a protocol that recommends a significantly longer period. This limited duration likely contributed to the lack of observable effects on the chemical composition of the insect meal and may have also played a role in the lower antioxidant activity recorded in the experimental groups.
Reviewer 2 Report
Comments and Suggestions for Authors
The manuscript presents the results of feeding trials with Tenebrio molitor larvae using brewers’ spent grain and carob flour (Ceratonia siliqua) in a scientifically sound and well-written manner. While the use of spent grain as a feed substrate for T. molitor has already been addressed in various studies, the investigation of carob flour as a dietary component appears to be novel and constitutes a meaningful contribution to the field.
The manuscript is coherent in structure, and the presentation of the data, as well as the conclusions drawn, are appropriate.
Nevertheless, the authors are strongly encouraged to clearly specify the type, processing method, and commercial form of spent grain utilised, as these factors can substantially affect the outcomes of insect feeding trials—both positively and negatively.
In addition, the inclusion of a more detailed characterisation of the sugar composition of C. siliqua and its digestibility or bioavailability to insects would significantly strengthen the manuscript. I recommend publication after minor revision.

Reviewer 3 Report
Comments and Suggestions for Authors
This manuscript addresses an important question in insect rearing sustainability by testing agro-industrial by-products (carob and BSG) as substrates for Tenebrio molitor. The experimental design is sound, and the manuscript is well written, but several clarifications, methodological justifications, and deeper discussions are needed before publication. Specific attention should be given to data interpretation, methodological transparency, and the link between feed composition and functional properties of insect meals.
Specific suggestions
line 44-46: Review sentence. Suggested: “Among the most pressing concerns is the projected 52% increase in global demand for animal protein by 2050, according to the FAO [2].”
line 46-47: Clarify that urgency is driven by sustainability constraints, not just demand increase.
line 61: fix missing space with reference: “...insect rearing [10].”
line 95-96: review sentence, it sounds like the larvae were used for hydration. Suggested: “Seven-week-old larvae, reared on wheat bran and hydrated with vegetables (potato and carrot)...”
line 105: Describe drying and grinding method for carob and BSG—temperature, duration, grinder type?
line 107: Explain why additional water was sprayed even with vegetable moisture. Was this standardised hydration or to support substrate mixing?
line 114: Clarify weight recording method—was this fresh or dry weight? Were larvae weighed collectively or individually?
line 126: you did not mention in the previous design description that the substrate weight was recorded. Add the methodology.
line 133-135: you used the weight of a 7 days-old larva (c.a. 0.2 g) to also estimate the survival rate of 14 days old larvae? It’s not very clear and this method does not seem accurate. Why did you not count the larvae? Automatic counting could have been done with pictures and a counting software, with insect counting parameters. Please justify or acknowledge this limitation.
line 138: A 1 mm sieve may not exclude small residues or faeces. Justify choice or discuss potential contamination.
line 139: drying what?
line 144: no reference for the NSC calculation? (even AOAC-style citation).
line 154: Clarify terms “substrate” (input feed) vs “meal” (processed larvae). These are sometimes conflated—state definitions early.
line 157: but without technical replicates this time?
line 207: Why was “replicate” not modelled as a random effect? Justify the statistical model, especially since trays were experimental units.
line 218: Clarify how growth rate (%) was calculated, it’s not described in methods. Add formula.
line 228: Cannot the survival rate scale adjusted to 100%?
line 248: “According to our findings” is misleading since you cite external data. Rephrase.
line 250: you did not discuss the differences in FCR?
line 270-272: no reference for this statement?
line 288-289: I did not really get the explanation here, was the drying process performed differently for the control group?
line 294: “may be due to the inclusion rates” — elaborate. What rates are optimal based on previous literature?
line 296-297: is the BSG’s profile from the reference 41? Maybe precise it.
line 309: Here you could further discuss the differences observed between the substrates and meals. You also measured them at different time points, but this is not really shown in the substrate and meal results. It would be interesting to further discuss how the composition differs between the substrate and mealine
line 320: revise “figure a” to Figure 2, a
line 326: same here, Figure 2, b
line 342: missing space with reference “metabolism[46-48]”
line 342-345: you show below that sequestration is unlikely as the phenol content was not higher in comparison to the control?
line 363 – 364: repetition of suggest and suggesting
line 368: reference?
line 369: review sentence. Regarding the inclusion of carob or brewer’s spent grain in the insect diet, it may reduce…
line 382: interesting; this could be expanded. What further experiments would you suggest to verify the antioxidant metabolism or bioavailability?
Additional Suggestions
- Figures and Tables: Make sure figure captions are self-explanatory and match citation format (e.g. "Figure 2a").
- Terminology: Standardise “flour” vs “meal” vs “extract” usage across the manuscript.
- Discussion: Expand on the ecological and economic implications. How feasible is by-product scaling in industrial settings?
Recommendation: Minor Revisions
The manuscript is well conceived, but certain aspects, especially around method transparency and statistical modelling require attention. The study is worth publishing after these improvements.
Best wishes.
